# Diverse Virulence Attributes of *Pantoea alfalfae* sp. nov. CQ10 Responsible for Bacterial Leaf Blight in Alfalfa Revealed by Genomic Analysis

**DOI:** 10.3390/ijms24098138

**Published:** 2023-05-02

**Authors:** Bo Yao, Rong Huang, Zhenfen Zhang, Shangli Shi

**Affiliations:** 1Key Laboratory of Grassland Ecosystem, Ministry of Education, College of Grassland Science, Gansu Agricultural University, Lanzhou 730070, China; 2Sino–U.S. Centers for Grazing Land Ecosystem Sustainability, Ministry of Science and Technology, Lanzhou 730070, China

**Keywords:** novel species, *Pantoea alfalfae* sp. nov., alfalfa, bacterial leaf blight, virulence, motility, biofilm

## Abstract

Alfalfa is widely grown worldwide for its excellent nutritional value. *Pantoea* species living in alfalfa seeds can easily spread over great distances with frequent trade. However, the pathogenic properties of this dangerous hitchhiker on alfalfa have not been evaluated. Here, we identified the taxonomic status of *Pantoea* strain CQ10 isolated from the interior of alfalfa seeds based on the whole genome sequence. The diverse virulence attributes of strain CQ10 during host infection were characterized through pathogenicity assays and functional and genomic analyses. We report that strain CQ10 belongs to a novel species in the genus *Pantoea*, which was phylogenetically close to *Pantoea vagans* and *Pantoea agglomerans*. Strain CQ10 caused bacterial leaf blight of alfalfa after inoculation from the roots. We found that strain CQ10 possesses a large number of pathogenic genes involved in shaping the virulence properties during bacteria–host interactions, including motility, biofilm, type VI secretion system, and nutrient acquisition. Compared with *P. vagans* and *P. agglomerans*, the unique virulence factors of strain CQ10 were mainly involved in motility and biofilm, which were confirmed by in vitro experiments. Taken together, our results suggest that strain CQ10 is the first *Pantoea* species to infect alfalfa, and it possesses diverse virulence attributes among which motility and biofilm may be the best weapons.

## 1. Introduction

Alfalfa (*Medicago sativa* L.) is a high–quality forage legume that is widely grown around the world and is favored by livestock for its nutritional properties rich in protein, vitamins, amino acids, and minerals [1]. Likewise, alfalfa sprouts are also widely consumed as ready–to–eat products by humans. These broad requirements will need to be met in the future by growing more high–yielding and healthy alfalfa plants. Bacterial disease is one of the constraint factors in alfalfa production. Bacterial root rot and leaf spot caused by *Pseudomonas viridiflava* and *Xanthomonas alfalfae* have been reported to be prevalent in alfalfa planting areas around the world [2,3]. *Pseudomonas syringae* pv. *syringae* is the causative pathogen of bacterial stem blight and causes 50% or more loss of alfalfa yield in Iran, Europe, Australia, and the United States [4]. The extent of risk of new or re-emerging bacterial diseases has increased as a result of increased trade of products between countries [5]. Seeds are an important part of the alfalfa trade commodity, which facilitates the long range spread of seed–borne bacteria such as *Erwinia persicina* [6]. The internal environment of the seeds provides a shelter for bacterial cells, which allows the bacteria to survive for long periods of time with seed storage. *Pseudomonas*, *Pantoea*, *Erwinia*, and *Cronobacter* are characterized as alfalfa seed endophytes that can survive in seeds for periods of more than five years [7,8]. Moreover, human pathogens can spread through alfalfa seeds and cause foodborne illnesses. For example, the seed–borne *Salmonella enterica* or *Escherichia coli* enter the sprouts during the germination process and then wait for the opportunity to infect when people eat the sprouts [9]. Therefore, extensive and intensive research on alfalfa seed–borne bacteria is of great significance to ensure that alfalfa can better serve human beings.

The genus *Pantoea* is a class of functionally diverse Gram–negative bacteria in the Enterobacteriaceae family. *Pantoea* species have a wide range of habitats and have been isolated from soil, water, plants, animals, and humans in the past [10]. Many species of *Pantoea* exhibit host diversity as pathogens. *P. ananatis* causes severe loss of pineapple, corn, rice, onion, and eucalyptus in plant hosts, and can cause bacteremia in humans [11,12]. *P. dispersa* and *P. eucrina* can cause diseases in plant hosts (rice and cotton) as well as bacteremia in humans [13,14,15,16]. In addition, the widely studied *P. agglomerans* have a greater pathogenic ability to infect a wide range of hosts in plants, animals, and humans [17]. This ability to cause disease in plant and animal hosts depends on the diverse virulence attributes of pathogens. Generally, motility plays an important role in the early phases of bacterial infection of plant and animal hosts, which mainly helps bacterial cells to adhere to the host and reach the preferred niches [18]. When bacteria reach infection sites within the host, the formation of biofilms is the key to successful colonization and persistent infection [19]. The effectors are then delivered into the host cell via the bacterial secretion system as the weapon to trigger the disease [20]. However, studies on how *Pantoea* bacteria coordinate various virulence factors in the process of infecting animal and plant hosts are still lacking.

In the past, the classification of many species in the genus *Pantoea* was inaccurate. We learned that seven species have been reassigned from *Pantoea* to the new genus of *Tatumella* and *Mixta* [21,22]. This suggests that traditional taxonomic methods are insufficient for accurate distinction and identification of *Pantoea* species. The marker genes such as 16S rRNA, *leuS*, *fusA*, *gyrB*, *rpoB*, and *rlpB* are often used to define *Pantoea* species and assess their evolutionary uniqueness, but these genes often exhibit higher sequence conservation than the genome average. Therefore, analysis based on a single or a set of generally conserved genes does not provide sufficient resolution for the classification and identification of *Pantoea* at the species level [23,24]. The complete whole genome sequence (WGS) provides new methods for better determining the evolutionary and taxonomic relationships of bacteria. The average nucleotide identity (ANI) analysis based on WGS has emerged as a reliable method for identifying and distinguishing closely related species. The ANI threshold range for species classification is generally 95–96% [25]. Digital DNA–DNA hybridization (dDDH) is another method for inferring distances between different genomes, usually with a species threshold of 70% [26].

We report here the isolation and identification of a novel species (strain CQ10) of the genus *Pantoea* isolated from alfalfa seeds by combing 16s rRNA and WGS analysis, which we propose to name “*Pantoea alfalfae* sp. nov.”. The results of the pathogenicity assay revealed that strain CQ10 is a new pathogen of bacterial leaf blight in alfalfa. Further genomic analysis revealed the pathogenic potential of strain CQ10 exploiting motility, biofilms, nutrient uptake, and type VI secretion systems to infect hosts. Overall, the findings of this study bring new insights into the pathogenic mechanisms of *Pantoea* species and the pathogenic potential of alfalfa seed–borne bacteria in grassland agro–ecosystems.

## 2. Results

### 2.1. Isolation of a Novel Pantoea Species

On day 2, we isolated a dominant strain CQ10 (approximately 10^5^ CFU/g) from the seeds of alfalfa, whose colonies showed a yellow color and appeared smooth, shiny, circular and convex on the LB agar, NB agar, and TSB agar plates after 48 h at 30 °C (Appendix A). The maximum diameter of single colony was about 0.25 cm, 0.35 cm, and 0.40 cm on NB agar, LB agar, and TSB agar plates, respectively (Appendix A). SEM characterized the strain CQ10 as short and rod–shaped. In addition, the cells were about 0.5 μM wide by 0.8–1.8 μM (Appendix A). The phylogenetic analysis based on 16S rRNA gene sequences indicated that strain CQ10 belongs to the genus *Pantoea* and formed a deep–branching lineage (Appendix A).

### 2.2. General Genome Features of Strain CQ10

The genome of strain CQ10 consists of a single circular chromosome and eight plasmids (Appendix A). The PROKKA pipeline annotated a total of 4522 genes in the CQ10 genome, which include 4330 coding gene sequences (CDS), 81 tRNAs, 22 rRNAs, one tmRNA, and 88 misc_RNAs (Figure 1A). Among these, 2513 (58.04%) coding genes were assigned to 23 clusters of orthologous groups (COG) functional categories (Figure 1B). The top five categories were amino acid transport and metabolism, carbohydrate transport and metabolism, translation, ribosomal structure and biogenesis, cell wall/membrane/envelope biogenesis, and energy production and conversion. Meanwhile, the last three groups were “extracellular structures”, “mobilome: prophages, transposons”, and “RNA processing and modification” (Appendix A).

### 2.3. Taxonomic Identification of Pantoea sp. Strain CQ10

The pairwise ANI and dDDH values between strain CQ10 and 29 species of the genus *Pantoea* ranged from 73.02% to 94.25%, and 19.9% to 58.2%, respectively (Appendix A; Appendix A). These values were below the proposed species boundary cut–off value of 95–96% ANI and 70% dDDH. The closest related species were *P. agglomerans* and *P. vagans*. Overall, the combination of 16S rRNA, ANI, and dDDH analyses showed that strain CQ10 belongs to a *Pantoea* novel species, we proposed the name *Pantoea alfalfae* sp. nov. for strain CQ10.

### 2.4. Biochemical Characteristics

The carbon source utilization and other biochemical tests of *P. alfalfae* sp. nov. were performed using API 20E and GEN Ⅲ microplates according to the manufacturer’s instructions. The detailed results are given in the species descriptions. *P. alfalfae* sp. nov. can be differentiated from closest *Pantoea* species by the ability to utilize amygdalina, D–arabitol, L–pyroglutamic acid, Tween 40, D–fucose, raffinose, and L–fucose (Table 1).

### 2.5. Alfalfa Bacterial Leaf Blight Caused by Strain CQ10

After inoculation with strain CQ10, the growth of alfalfa plants was significantly inhibited (Figure 2A). Compared with the CK group, the alfalfa leaves showed severe wilting, chlorosis, and patchy necrosis by 21 days in the CQ10 group (Figure 2B,C). Analysis of the photosynthetic physiology of leaves showed that the chlorophyll a (Chla), chlorophyll b (Chlb), and carotenoid (Car) contents of the CQ10 group were significantly reduced, and the total chlorophyll content was reduced by 89.03% compared with the CK group (Figure 2D). Meanwhile, CQ10–treated plants significantly decreased the net photosynthesis rate (Pn), stomatal conductance (Gs), and transpiration rate (Tr) compared with the CK group, while the intercellular CO_2_ concentration (Ci) was significantly increased (Figure 2E–H). Strain CQ10 also had a significant negative effect on alfalfa seed germination and biomass (Figure 2I–K).

Notably, strain CQ10 demonstrated a strong inhibitory effect on the growth of alfalfa primary roots (Figure 3A). In addition, plant length in the CQ10 group was significantly less compared with that of the CK group, which was mainly due to the strong inhibition of root growth (Figure 3B). The analysis of root morphology indicated that, compared with non-inoculated controls, the total root surface area and root volume of alfalfa in the CQ10 group were decreased markedly, while the average root diameter was significantly increased (Figure 3C). This was consistent with the observed phenomenon of root thickening. Furthermore, to validate Koch’s postulates, strain CQ10 was re-isolated from the leaves, stems and roots of the CQ10 group. Altogether, the results obtained indicate that the strain CQ10 was a causative agent of bacterial leaf blight of alfalfa plants, and it severely inhibited the growth of alfalfa roots and caused leaf disease.

### 2.6. Changes in the Antioxidative System of Alfalfa Plants

The change of antioxidant defense system can reflect the oxidative damage of pathogenic bacteria to plants. To understand the response of the antioxidant system of alfalfa plants under CQ10 stress, the activities of catalase (CAT, EC 1.11.1.6), peroxidase (POD, EC 1.11.1.7), ascorbate peroxidase (APX, EC 1.11.1.11), superoxide dismutase (SOD, EC 1.15.1.1), and malondialdehyde (MDA) contents were measured in roots, stems, and leaves at days 21, respectively (Figure 4A–E). In the roots, inoculation of alfalfa plants with strain CQ10 resulted in a statistically significant increase in SOD, APX, and POD activities compared with non-inoculated controls, while CAT activity and MDA content were decreased significantly. In the stems of alfalfa, all antioxidant enzyme activities in plants inoculated with strain CQ10 were significantly increased compared with the CK group. However, the MDA content in the CQ10 group was slightly lower than the non-inoculated control and there were no significant differences. The opposite results were observed in the alfalfa leaves compared with those in the roots. When compared with the CK group, the SOD and APX activities in alfalfa leaves inoculated with strain CQ10 were significantly decreased while CAT activity increased. Meanwhile, the CQ10 treatment significantly increased the activity of POD in the leaves of alfalfa by 38.2% and MDA content decreased by 48.6%.

### 2.7. Analyses of Defense–Related Enzymes in Alfalfa Hosts

The activity level of plant defense enzymes is related to host resistance and virulence of pathogenic bacteria. Here, the activities of phenylalanine ammonia-lyase (PAL, EC 4.3.1.5) and polyphenol oxidase (PPO, EC 1.10.3.1) were determined to understand the effect of induced resistance on alfalfa plants by strain CQ10 at day 21 (Figure 4F,G). When compared with the CK group, the PAL activity in stems and leaves of alfalfa inoculated with strain CQ10 significantly decreased by 16.2% and 32.2%, respectively. However, there was no significant difference in PAL activity between the CQ10 group and the non-inoculated control. The PPO activity increased in the leaves at 21 days after inoculation with strain CQ10 compared with the CK group, while it decreased in the roots. Meanwhile, the activity of the PPO enzyme in the CQ10 group was slightly lower than in the CK group (no significance).

### 2.8. Effect of Strain CQ10 Inoculation on Soluble Sugar and Soluble Protein in Alfalfa Plants

As shown in Figure 4H, the inoculation of alfalfa plants with strain CQ10 resulted in a statistically significant decrease of the soluble sugar (SS) content in the root (43.1% decrease), stem (37.6% decrease), and leaf (25.3% decrease) compared with non-inoculated control. The opposite result was observed for the content of soluble protein (SP) (Figure 4I). When compared with the CK group, the soluble protein content under the inoculated treatment was significantly increased. Numerically, soluble protein content in the CQ10 group increased in the roots, stems, and leaves by 40.4%, 11.9%, and 30.0%, respectively, compared with the CK group.

### 2.9. Correlation between Nine Physiological Parameters

To explore the relationship between antioxidant defense systems, defense–related enzymes, and soluble species, the correlation among nine physiological indicators was analyzed using Spearman’s correlation test. In the roots (Figure 5A), SOD activity was significantly positively correlated with APX activities and SP contents but negatively correlated with PAL activities and MDA contents. CAT activities had a significant positive correlation with PPO activities and SS contents. APX and POD activities were significantly positively correlated with the antioxidant enzymes. Meanwhile, there was a significantly negative correlation between MDA and SP contents, and a positive correlation with PAL activity. Interestingly, we found significant pairwise correlations between SOD, MDA, SP, and PAL (Figure 5B). In stems (Figure 5C), SOD activity showed positive and highly significant correlations with POD enzyme activity and negative correlations with PPO, PAL activities, and MDA content. CAT activity showed a significant positive correlation with SP content, while there was a significant negative correlation with PPO and PAL activities and MDA content. Meanwhile, a significant positive correlation was found between POD activity, APX activity, and SP content, and between SP content and APX activity. However, POD activity and SS content had a significant negative correlation with PPO activity and SP content, respectively. In further analysis, we found significant pairwise correlations between SOD, POD, and PPO enzyme activities (Figure 5D). As is shown in Figure 5E, SOD activity had a significant positive correlation with PAL activity and SS content in the alfalfa leaves, and a negative correlation with CAT and POD activities. At the same time, a significant positive correlation between CAT activity and POD activity was observed in the leaves; however, PAL activity and SS content had a significant negative correlation with CAT activity. Moreover, APX activity showed a significant positive correlation with MDA content, and there was a negative correlation between PPO activity and SP content. POD activity showed a significant negative correlation with PAL activity and SS content. MDA content was highly negatively correlated with SP content. Apart from this, PAL activity was significantly and positively associated with SS content. Of note, there was a high pairwise correlation between SOD, POD, CAT, PAL, and SS. Besides this, APX, MDA, and SP also had significant pairwise correlations (Figure 5F).

To evaluate the correlation in different parts of alfalfa plants, we further analyzed the nine physiological indices in the root, stem, and leaf of alfalfa. We found that SOD activity was significantly correlated with PAL activity in alfalfa roots, stems, and leaves (Appendix A, highlighted in red font). Otherwise, SOD activity had a significant correlation with POD activity in stems and leaves. Meanwhile, there was a significant indirect correlation between SOD activity and POD activity in roots because SOD was significantly correlated with APX, while APX was correlated with POD in roots. Thus, SOD activity had a significant correlation with POD activity in roots, stems, and leaves (Appendix A, highlighted in blue font). Likewise, evidence for a significant correlation between CAT activity and SS content was observed in alfalfa roots, stems, and leaves (Appendix A, highlighted in green font). Moreover, as shown in Appendix A, we also found that MDA content had a significant correlation with SP content in roots, stems, and leaves (highlighted in orange font).

### 2.10. Pan–Genome Analysis of Pantoea Strains

To investigate differences between strain CQ10 and other closely related strains, we implemented a PGAP of eight genomes (ANI value > 90%). A whole genome comparison based on coding sequences revealed a core genome shared by eight *Pantoea* strains of 3412 (55.9% of total pan genome) orthologous groups (OGs) and a pan genome of 6104 OGs (Figure 6A). Moreover, we found that the number of *Pantoea* strain–unique OGs varied from 72 to 287, with strain CQ10 having 193 (3.2% of total pan genome) unique OGs (Figure 6A). Functional annotation of genes in each orthologous group was performed based on the COG database (Figure 6B). We found that most core genes were classified in “Function unknown (S)”, “Amino acid transport and metabolism (E)”, “Inorganic ion transport and metabolism (P)”, “Transcription (K)”, “Carbohydrate transport and metabolism (G)”, “Cell wall/membrane/envelope biogenesis (M)”, “Energy production and conversion (C)”, “Coenzyme transport and metabolism (H)”, and “Translation, ribosomal structure and biogenesis (J)” (Figure 6B; Appendix A). Altogether, 43.5%, 20.0%, 19.4%, and 17.0% of these core genes were, respectively, assigned to the “Metabolism”, “Cellular processes and signaling”, “Poorly characterized”, and “Information storage and processing” categories of COG (Appendix A).

Furthermore, a total of 758 (54.3% of 1395 strain–unique genes) genes in eight *Pantoea* strains were assigned to “Poorly characterized” (31.1%), “Metabolism” (28.0%), “Cellular process and signaling” (22.8%), and “Information storage and processing” (18.1%) (Figure 6A,B; Appendix A). Among these COG categories, we found that unique genes of strain CQ10 contained a high abundance of genes involved in “Metabolism” (50%), which was a greater proportion than that found in the other seven strains. Specifically, most unique genes of strain CQ10 were categorized into “Inorganic ion transport and metabolism (P)”, “Coenzyme transport and metabolism (H)”, “Secondary metabolites biosynthesis, transport and catabolism (Q)”, “Posttranslational modification, protein turnover, chaperones (O)”, “Cell motility (N)”, and “Cell cycle control, cell division, chromosome partitioning (D)”, and it also can be seen that these proportion were greater in strain CQ10 compared with other seven strains.

### 2.11. Comparative and Functional Genome Analyses

To further understand the genomics similarities and differences between strain CQ10 and closely related *Pantoea* strains, we performed a whole–genome collinearity analysis. The results showed high collinearity between strain CQ10 and the other six *Pantoea* strains (Figure 7A). Specifically, the collinearity percentages between strain CQ10 chromosome and three *P. vagans* strains were higher than between strain CQ10 chromosome and three *P. agglomerans* strains (Appendix A). Remarkably, we found that there was no collinearity between strain CQ10 plasmid p7 sequence and genomes of all six *Pantoea* strains.

Furthermore, the virulence factors in strain CQ10 and the other six *Pantoea* strains were identified by aligning CDS sequences to the virulence factors in the VFDB. A total of 432 virulence factors (VFs) were detected within the seven *Pantoea* strains. Among them, 263 common VFs (60.9%) were shared by all seven strains (Appendix A). For strain CQ10, the top five common VFs with a larger number of encoding genes were “peritrichous flagella (motility)”, “pyoverdine (iron acquisition)”, “HitABC (iron acquisition)”, “FbpABC (iron acquisition)”, and “CdpA (regulation)”, respectively. Moreover, we further identified strain–unique virulence factors (Appendix A). We found that strain CQ10 had the greatest number of unique VFs. Among these unique VFs, the highest number of genes were involved in biofilm (46.7%), followed by motility (26.7%), immune modulation (13.3%), adherence (6.7%), and antimicrobial activity/competitive advantage (6.7%) processes (Figure 7A; Appendix A).

### 2.12. Biofilm Formation and Motility of Strain CQ10 In Vitro

As shown in Figure 7B, strain CQ10 was a strong biofilm producer. The biofilm formation ability of the strains at 23 °C and 37 °C was significantly lower than the optimum growth temperature of 30 °C, but there was no significant difference between 23 °C and 37 °C. Next, we examined the bacterial motility and also found that strain CQ10 had the highest motility at the optimum temperature of 30 °C, and there was no significant difference between 23 °C and 37 °C (Figure 7C).

### 2.13. Analysis of Virulence Attribute of Strain CQ10

Pathogen–host interactions (PHI) analysis of the genome sequence of strain CQ10 resulted in the functional annotation of 1603 CDS. We found that genes involved in the function of increased virulence and effector were 36.5% of the total annotated CDS. Among them, strain CQ10 harbored genes encoding different virulence factors associated with plant disease, and the specific results were presented in Appendix A. Importantly, we found a gene cluster on the chromosome of strain CQ10 that encodes a type VI secretion system (T6SS) (Figure 8A).

As shown in Appendix A, we identified 111 carbohydrate–active enzymes (CAZymes) in the genome of strain CQ10. GlycosylTransferases (GTs) were the largest family and account for 45.0% of all annotated CAZymes. Glycoside Hydrolases (GHs) was the second most abundant family (41.4%), followed by Carbohydrate–Binding Modules (CBMs, 7.2%), Auxiliary Activities (AAs, 3.6%), and Carbohydrate Esterases (CEs, 2.7%). Specifically, GT2 (30.0% of all GTs), GT4 (14.0%), and GT9 (12.0%) were among the most predominant in the GTs family. GH13 (15.2% of all GHs), GH1 (13.0%), and GH23 (10.9%) were among the most predominant families in the GHs. Notably, we found a set of genes involved in sucrose metabolism on the plasmid of strain CQ10 (Figure 8B).

Furthermore, the genome of strain CQ10 was searched against the antiSMASH database (Appendix A). Five and three biosynthetic gene clusters (BGCs) were found in the chromosome and plasmid of strain CQ10, respectively. Among them, the genes involved in the biosynthesis of bacillibactin, desferrioxamine E, and carotenoids in strain CQ10 were most similar to the gene arrangement of the corresponding BGCs in the minimum information about a biosynthetic gene cluster (MIBiG) database (Figure 8C–E).

## 3. Discussion

Since the genus *Pantoea* was first described in 1989, many different members were identified as *Pantoea* species. However, some species were misassigned to *Pantoea* due to limitations of traditional bacterial taxonomy [27]. In this study, we proposed strain CQ10 as a new species, named *Pantoea alfalfae* sp. nov., in the genus *Pantoea* based on whole genome data analysis results [25]. Meanwhile, we need to illustrate that the whole genome sequence of strain CQ10 has been submitted to the NCBI database (ASM1988020v1) but the strain name in the NCBI database (*Pantoea agglomerans* CQ10) was inaccurate. Many *Pantoea* species are phytopathogenic bacteria with a wide host range; however, there are no reports of *Pantoea* causing alfalfa disease [28]. In this work, we characterized the strain CQ10 as a novel pathogen of alfalfa bacterial leaf blight, which enriches the list containing only *Xanthomonas alfalfae* and *Erwinia persicina* as pathogens of alfalfa bacterial leaf blight [3,6]. Further genomic analysis revealed that strain CQ10 possesses diverse virulence attributes, which may support a broad pathogenic potential, similarly to its close relative *P. agglomerans*.

Motility is a very important virulence factor for pathogenic bacteria, especially in the early phases of infection [18]. We found that the strain CQ10 possessed a large number of virulence genes involved in motility compared with *P. agglomerans* and *P. vagans* (Appendix A). In particular, the strain–unique virulence factors contained on the plasmid of strain CQ10 were mainly involved in bacterial motility (Figure 7A; Appendix A), which may help strain CQ10 acquire stronger motility than closely related *Pantoea* species. In vitro motility experiments revealed that strain CQ10 had no significant difference in motility at 23 °C and 37 °C (Figure 7C), which indicated that strain CQ10 may have the same motility ability during the interaction with animal and plant hosts. The flagella–mediated motility has already been shown to contribute bacterial cells to spread throughout the plant and reach the optimal host site [29,30], suggesting that motility may be a key factor for strain CQ10 to reach alfalfa leaves. The apparent changes when the bacteria successfully reach the optimal host site is the cessation of flagella–mediated motility, and then, a bacterial biofilm is usually formed [30,31]. Bacterial biofilm formation on host tissues is a key strategy contributing to pathogenicity [32]. Compared with *P. agglomerans* and *P. vagans*, most of the strain–unique virulence factors of CQ10 were involved in biofilm formation (Figure 7A). In vitro experiments also confirmed that strain CQ10 was a strong biofilm former (Figure 7B). Biofilm formation during strain CQ10–host(s) interactions may support enhanced nutrient uptake and resistance to host defense compounds by strain cells at the host site [33].

The type VI secretion system (T6SS) plays an important role in bacterial pathogenicity, which may be due to a direct role through the infection of host cells by secreted effectors or an indirect role through its mediated antibacterial competition, motility and, biofilm formation [34,35]. *P. agglomerans*, *P. vagans*, and *P. ananatis* contain T6SS gene clusters, and the T6SS of *P. ananatis* has been proven to play an important role in pathogenicity and bacterial competition [20,36]. We also found components involved in the synthesis of the T6SS apparatus in the genome of strain CQ10, including TssA–TssG, TssJ–TssM, Hcp, VgrG, and ClpV (Figure 8A). Among them, VgrG forms a cell–puncturing tip and Hcp forms a tail–tube structure assembled at the bottom of VgrG, which is responsible for delivering effectors into cells of other bacteria. TssB and TssC form a helical sheath, which is responsible for providing energy for effector translocation. ClpV primarily disassembles the contracted sheath and recycles its components for subsequent secretion events. Interestingly, the structural components VgrG (valine–glycine repeat protein G) and Hcp (haemolysin co-regulated protein) are also important T6SS effectors [35].

Efficient access to nutrients determines the survival and infection of pathogens in the host. The soluble sugar in the plant host is the carbon source of the pathogenic bacteria. We observed that the soluble sugar of alfalfa, consisting of sucrose (the most abundant), fructose, glucose, maltose, and trehalose, in the strain CQ10 treatment group was significantly lower than that in the control group (Figure 4H), which may be caused by the consumption of sugar by a large number of carbohydrate active enzymes encoded by the strain CQ10 (Appendix A) [37]. Notably, there was a gene cluster involved in sucrose metabolism on the plasmid of strain CQ10 (Figure 8B), wherein, the catabolite repressor/activator (Cra) encoded by the *cra* gene is a regulator involved in global carbon metabolism and plays an important role in sucrose metabolism [38]. The *sacX* and *scrY* genes are mainly involved in the transmembrane transport of sucrose, and *scrB* and *scrK* are mainly involved in the hydrolysis of sucrose [39]. We speculate that this gene cluster may have been acquired through a horizontal gene transfer (HGT) event to adapt and utilize the high content of sucrose in alfalfa.

Iron acquisition strategies play vital roles in infection by plant pathogenic bacteria [40]. In this work, the reduction in chlorophyll and photosynthesis may be related to the iron limitation caused by strain CQ10 (Figure 2D–H), because the strain CQ10 genome was rich in virulence factors involved in iron acquisition and transport, including HitABC, FbpABC, pyoverdine, bacillibactin, desferrioxamine E (Figure 6B, Figure 8C,D and Appendix A). Generally, once ferric iron is extracted by siderophores (pyoverdine, bacillibactin, and desferrioxamine E) from transferrin, lactoferrin, or the host environment, the iron will be transported into the pathogen via the iron transport system (HitABC and FbpABC) [41,42,43]. Interestingly, bacillibactin is mainly found in bacteria of the genus *Bacillus* and has not been reported in *Pantoea* [44]. Compared with the bacillibactin biosynthesis gene cluster of *Bacillus velezensis* FZB42, we found that the gene arrangement in the genome of the strain CQ10 was rearranged, and many additional biosynthetic genes and transport–related genes were added to make the structure of the gene cluster more complicated (Figure 8C). Moreover, desferrioxamine E located on the plasmid of the strain CQ10 was highly similar to that in *P. agglomerans* (Figure 8D), which may be originated from *Erwinia* through HGT to enhance the iron acquisition of the bacteria [45].

In conclusion, *P. alfalfae* sp. nov. CQ10 caused bacterial leaf blight of alfalfa depending on diverse virulence attributes, mainly including motility, biofilm, T6SS, and nutrient acquisition. These virulence attributes are common strategies used by *Pantoea* species to infect animal and plant hosts, but comparative genomic analysis revealed that strain CQ10 may have enhanced motility and biofilm–forming abilities. Given the diverse pathogenic properties of *Pantoea* species, future work needs to investigate how strain CQ10 coordinates the effects of various virulence factors when infecting hosts and whether strain CQ10 is pathogenic to other plant and animal hosts in grassland agro–ecosystems.

*Pantoea alfalfae* [al.fal′fae. L. n. *alfalfae* of alfalfa (*Medicago sativa* L.), referring to the host plant from which the first strains where isolated].

Cells are Gram–strain negative, short rod shaped (0.5 × 0.8–1.8 μM), motile and non-spore–forming. Colonies of strain CQ10^T^ were yellow, round, smooth and convex with approximately 0.25 cm, 0.35 cm, and 0.40 cm in diameter after culture at 30 °C for 48 h on NB agar, LB agar, and TSB agar plates, respectively. Strain CQ10 was negative for oxidase, but positive for reduction of nitrate to nitrite and for Voges–Prokauer reaction. Gelatinase is produced by strain CQ10. In addition, β–galactosidase, double–arginine hydrolase, lysine decarboxylase, ornithine decarboxylase, tryptophan dehydrogenase, indole, H_2_S, and urease are not produced. Acid is produced from glucose, mannitol, inositol, L–rhamnose, sucrose, amygdalina, and arabinose. The following compounds are utilized as carbon sources: D–arabitol, L–pyroglutamic acid, D–fucose, L–fucose, D–fructose, D–mannose, D–maltose, D–trehalose, glycerol, L–alanine, L–histidine. Strain CQ10 is unable to utilize D–raffinose and D–turanose as carbon sources. The DNA G + C content is 59.9 mol%.

The type strain is CQ10^T^, isolated from alfalfa seeds in China. The whole genome sequence of strain CQ10^T^ can be found at https://ftp.ncbi.nlm.nih.gov/genomes/all/GCF/019/880/205/GCF_019880205.1_ASM1988020v1/ (accessed on 22 March 2023).

## 4. Materials and Methods

### 4.1. Isolation and Characterization of Strain CQ10

For the isolation of endophytes, surface sterilization of alfalfa seeds was performed as described by Howard and Hutcheson [46] with appropriate modifications. Alfalfa seeds (1.0 g) were submerged in a 10% sodium hypochlorite solution for 15 min with occasional vortexing and washed three times with sterile distilled water. Then, the seeds were homogenized in 10 mL of sterile water. After standing for 20 min, 1 mL of the supernatant was diluted and 100 μL of it was evenly spread on the LA plate. Plates were incubated at 30 °C for the growth of isolates. The isolate (strain CQ10) was characterized by its ability to grow in NA, LA, and TSA plates for 48 h at 30 °C. Scanning electron microscopy (SEM) imaging was carried out by HITACHI Regulus 8100 (Hitachi Ltd., Tokyo, Japan). Phenotypic characteristics of strain CQ10 and closely related *Pantoea* species were performed using the API 20E (BioMérieux) and GEN Ⅲ microplates (Biolog). *Pantoea agglomerans* JCM 1236 and *Pantoea brenneri* LMG 5343 were purchased from Guangdong Microbial Culture Collection Center. *Pantoea vagans* DSM 23078 was purchased from Mingzhoubio (Ningbo, China).

### 4.2. DNA Extraction and 16S rRNA Gene Sequencing

A single colony of strain CQ10 was picked and incubated in 100 mL NB, which were grown at 30 °C and 160 rpm until OD600 of 1.0. The bacterial cells were harvested by centrifugation (5000 rpm, 15 min). Then, genomic DNA was extracted using a Bacterial Genomic DNA Extraction Kit (TIANGEN, Beijing, China). The 16S rRNA gene was amplified by PCR using the universal primers 27F and 1492R [47]. The PCR products were tested using 1% agarose gel electrophoresis and then sequenced by Sangon Biotech Company (Shanghai, China). Furthermore, the 16S rRNA gene sequence of strain CQ10 was compared to others in the GenBank database using the BLAST program to determine phylogenetic affiliations. The MEGA 11 program was used to align similar sequences (ClustalW algorithm) and created the phylogenetic tree [48].

### 4.3. Whole Genome Sequencing, Assembly, and Annotation

Genomic DNA (extracted as described previously) concentration and purity were quantified using NanoDrop 2500. The WGS of strain CQ10 was conducted using both Illumina and Nanopore methods by Benagen Tech Solution Co., Ltd. (Wuhan, China). After sequencing, a total of 1,333,116,600 bp (Illumina platform) and 2,059,427,800 bp (Nanopore platform) of clean reads were obtained from the raw reads by removing the connector, short fragments and low–quality sequences [49]. Then, the filtered reads were used for assembling by Unicycler version 3 (https://github.com/rrwick/Unicycler, accessed on 8 May 2022). Finally, the draft genome assembly error correction was performed using the Pilon software based on Illumina data [50]. The circular maps with gapless were generated and drawn using the CGView sever. The WGS of strain CQ10 was annotated using Prokka software [51]. In addition, each predicted CDS was annotated based on BLAST against the COG database. Genes associated with virulence were identified by screening whole genome sequences against the Virulence Factors Database (VFDB) and Pathogen–Host Interaction Database (PHI–base) [52,53]. Genes associated with carbohydrate–active enzyme families were detected using the dbCAN database [54]. Genes involved in secondary metabolism were discovered by conducting antiSMASH analysis [55]. The whole genome sequences of strain CQ10 were deposited in GenBank (accession numbers: CP082292–CP082300).

### 4.4. Phylogenomic Comparison

To further determine the taxonomic affiliation of strain CQ10, ANI and dDDH values between whole genomes of strain CQ10 and other closely related sequenced *Pantoea* strains were calculated using JSpeciesWS [56] and Genome–to–Genome Distance Calculator (GGDC) [57]. The WGS of *Pantoea* strains were obtained from the NCBI Genome database (https://www.ncbi.nlm.nih.gov/data-hub/taxonomy/53335/, accessed on 11 June 2022). ANI values of 95% and dDDH values of 70% were used as the boundary for species delineation. TBtools software (https://github.com/CJ-Chen/TBtools, accessed on 25 June 2022) was used to generate the heatmap.

### 4.5. Alfalfa Plant Inoculation Assay

The bacterial cells of strain CQ10 from 24 h cultures were harvested using centrifugation (5000 rpm, 5 min) and resuspended in sterile water. Then, a CQ10 inoculum suspension (10^9^ CFU/mL) was prepared. The healthy and full seeds of alfalfa (provided by the Sino–US Centers for Grazing Land Ecosystem Sustainability) were surface sterilized by submerging the seeds for 2 min in 75% (*v*/*v*) ethanol solution followed by 10 min in a 5% (*v*/*v*) sodium hypochlorite solution. Finally, alfalfa seeds were washed five times with sterile distilled water. The growth bottles were used for alfalfa plant inoculation experiments [6]. Briefly, CQ10 inoculum suspension (200 mL/bottle) was added to the growth bottles. For the control group (CK), an equal volume of sterile water was loaded. Then, alfalfa seeds (25 seeds/bottle) were placed at equidistant locations (seed spacing = 1 cm) in the germination bed of the growth bottle. The whole procedure was conducted in a sterile environment. After inoculation, all alfalfa plants were grown in a growth chamber at 23/20 °C, 16–h light/8–h dark (day/night). After inoculation for 21 days, the alfalfa plants were harvested and basic growth parameters including germination rate (GR%), germination potential (GP%), germination index (GI), plant height, fresh and dry weights, and leaf chlorophyll content were measured. In addition, the root morphological traits were analyzed using an Epson Expression 12000XL root system scanning analyzer (Epson, Nagano, Japan) and WinRHIZO Basic 2013 (Instruments Regent Inc., Sainte-Foy, QC, Canada).

### 4.6. Alfalfa Leaf Gas Exchange Measurements

A portable photosynthesis system (GFS–3000, Heinz Walz GmbH, Effeltrich, Germany) was used to measure Pn, Gs, Ci, and Tr at 1200 µmol/m^2^/s photosynthesis photon flux density (PPFD) and 380 µmol/m^2^/s CO_2_. Measurements were performed under light–saturated conditions (9:00 a.m. to 11:00 a.m.) on day 21. Then, the gas exchange parameters (Pn, Ci, Gs, and Tr) were calculated using equations developed by Von Caemmerer and Farquhar [58].

### 4.7. Measurement of Various Physiological Indices

Samples (root, stem, and leaf) were collected 21 days after the inoculation. Then, CAT, POD, APX, SOD, PAL, PPO, MDA, SP, and SS were determined according to the protocol of assay kits (Michy Biology, Suzhou, China). Briefly, 0.1 g plant samples were added into 1 mL of extraction buffer and ground into homogenate, then centrifuged at 12,000 rpm for 10 min at 4 °C. The supernatants were subsequently collected and used for further determination.

### 4.8. Pan–Genome Analysis

We performed a pan–genomic analysis to understand the genomic difference between strain CQ10 and other closely related *Pantoea* strains, which refer to *Pantoea* species with ANI values greater than 90% (Appendix A). The pan–genome calculation was carried out using the PGAP pipeline [59]. Orthologous gene clusters among these strains were examined using the GeneFamily (GF) method with default parameters. Moreover, the functions of core and unique genes were annotated using the COG database.

### 4.9. Genome Comparative Analysis

The whole genome sequences (complete genome level) were used as queries in the BLAST program to identify collinear regions in strain CQ10, three *P. agglomerans* strains, and three *P. vagans* strains. The results of collinearity among seven *Pantoea* strains were visualized by the circos program of TBtools software [60].

### 4.10. Biofilm and Motility Assays

Biofilm assays were performed as described with few modifications [61,62]. Briefly, we incubated polypropylene tubes containing suspensions of strain CQ10 at 23 °C, 30 °C, and 37 °C for 48 h. Then, 0.1% crystal violet (*w*/*v*) was used to stain biofilm cells and then dissolved in absolute ethanol. Quantitative analysis was performed by measuring absorbance at 595 nm. LB broth was used as a negative control. Bacterial strains were classified according to the cut–off optical density (ODc) defined as three standard deviations above the mean OD of the negative control: non-biofilm formers (OD ≤ ODc); weak biofilm formers (ODc < OD ≤ 2ODc); moderate biofilm formers (2ODc < OD ≤ 4ODc); and strong biofilm formers (OD > 4ODc). In order to test the motility, the strain CQ10 suspension was stabbed into the center of the LB plate containing 0.3% (*w*/*v*) agar. Plates were incubated at 23 °C, 30 °C, and 37 °C for 48 h and bacterial motility were observed [62].

### 4.11. Statistical Analysis

The statistical analysis was conducted using R 4.2 and SPSS 26.0 software. All correlation was calculated using Spearman’s correlation analysis. The circular map and heatmap were generated using Tbtools software. Venn diagram was drawn by using the OmicShare platform (https://www.omicshare.com/, accessed on 18 July 2022).

## Figures and Tables

**Figure 1 ijms-24-08138-f001:**
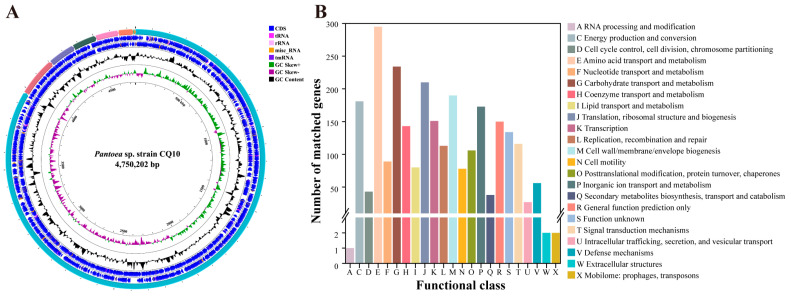
Genome analysis of strain CQ10. (**A**) Circular map and the circles from the outside to the center: track1, CQ10 chromosome, and plasmid contigs; tracks 2 and 3, coding sequence (CDS), RNA genes on forward and reverse strands, respectively; track 4, GC content; track 5, GC skew; track 6, scale of genome size. (**B**) Cluster of orthologous groups (COG) analysis of CQ10 genome.

**Figure 2 ijms-24-08138-f002:**
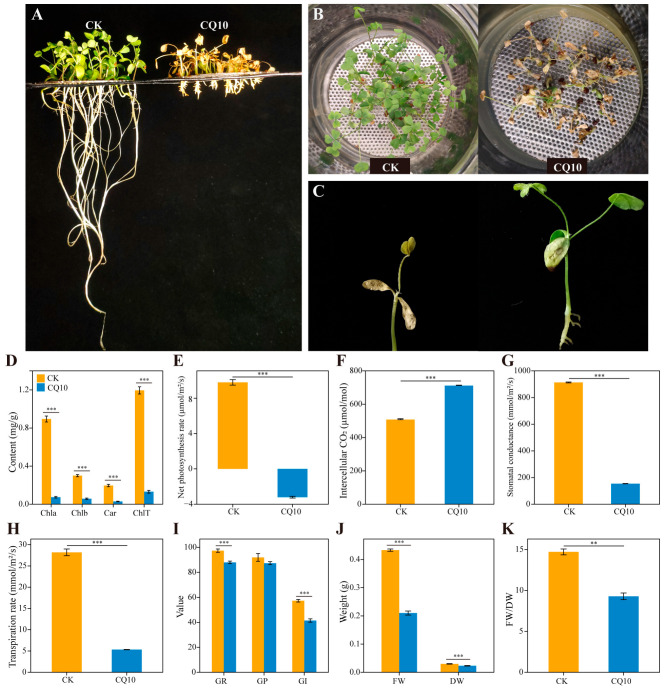
Alfalfa phenotypes and growth indicators at 21 days after inoculation with strain CQ10. Phenotypic comparison between CK and CQ10 groups: (**A**) front view and (**B**) top view. (**C**) Leaf spots, wilting, and necrosis symptoms on alfalfa leaves caused by CQ10. (**D**) Differences in the chlorophyll content of alfalfa leave in the CK and CQ10 (Welch’s *t*–test). Chla, chlorophyll a; Chlb, chlorophyll b; Car, carotenoid; ChlT, total chlorophyll. (**E**) Net photosynthesis rate. (**F**) Intercellular CO_2_ concentration. (**G**) Stomatal conductance. (**H**) Transpiration rate. Independent sample *t*–test. (**I**) Comparison of germination between CK and CQ10 (Welch’s *t*–test). GR (%), germination rate; GP (%), germination potential; GI, germination index. (**J**) Fresh weight (FW) and dry weight (DW) changes between CK and CQ10 groups (Independent sample *t*–test). (**K**) FW/DW: the fresh weight/dry ratio (Mann–Whitney test). **, *p* < 0.01; ***, *p* < 0.001; *n* = 6; mean ± SE.

**Figure 3 ijms-24-08138-f003:**
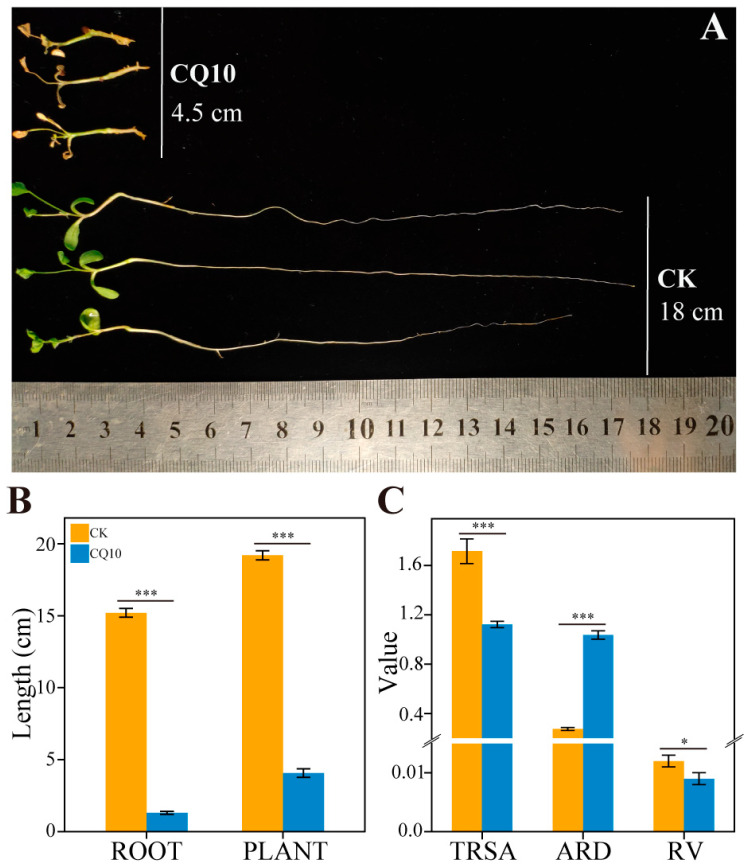
Strain CQ10 inhibited alfalfa root elongation and growth. (**A**) The photograph was taken at 21 days post–inoculation. (**B**) Comparison of root and plant lengths between CK and CQ10 groups (*n* = 9). (**C**) Differences in the alfalfa root growth indicators between CK and CQ10 groups (*n* = 6). TRSA, total root surface area (cm^2^); ARD, average root diameter (mm); RV, root volume (cm^3^). Independent sample *t*–test. *, *p* < 0.05; ***, *p* < 0.001; mean ± SE.

**Figure 4 ijms-24-08138-f004:**
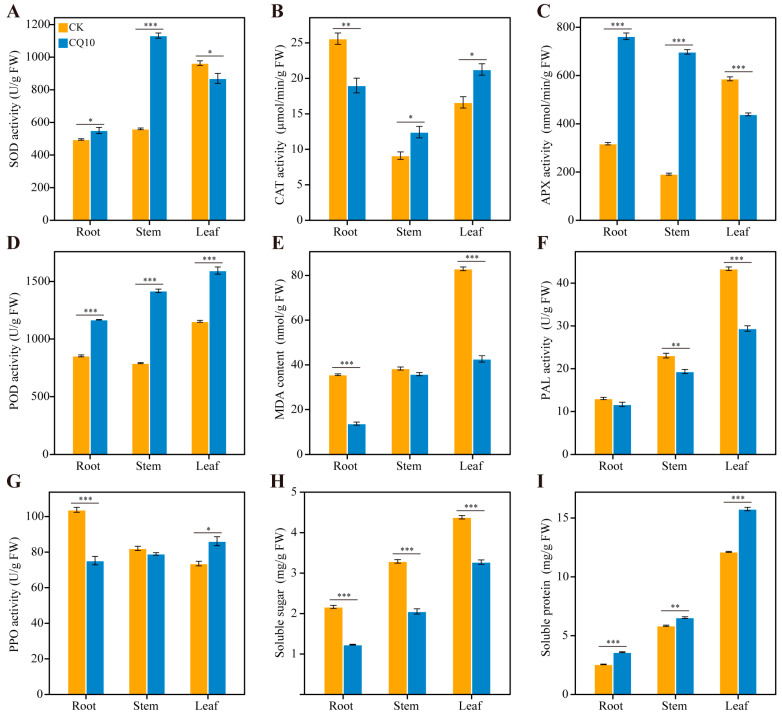
Comparison of nine physiological parameters between CK and CQ10 groups in different plant tissues after 21 days of inoculation with CQ10. (**A**) superoxide dismutase (SOD) activity. (**B**) catalase (CAT) activity. (**C**) ascorbate peroxidase (APX) activity. (**D**) peroxidase (POD) activity. (**E**) malondialdehyde (MDA) content. (**F**) phenylalanine ammonia lyase (PAL) activity. (**G**) polyphenol oxidase (PPO) activity. (**H**) soluble sugar. (**I**) soluble protein. FW, fresh weight. Independent sample *t*–test. *, *p* < 0.05; **, *p* < 0.01; ***, *p* < 0.001; *n* = 3; mean ± SE.

**Figure 5 ijms-24-08138-f005:**
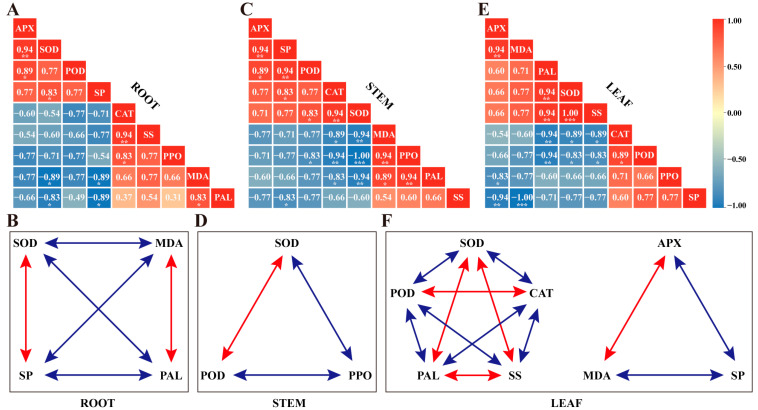
Spearman’s correlation between nine physiological indicators. Heatmap and pairwise correlation diagram in roots (**A**,**B**), stems (**C**,**D**), and leaves (**E**,**F**). The values within heatmap boxes represent the correlation coefficients (r values) and the asterisk represents a significant difference (*, *p* < 0.05; **, *p* < 0.01; ***, *p* < 0.001). The red color indicates a positive (0.00 < *r* < 1.00) correlation while blue a negative (−1.00 < *r* < 0.00) correlation. APX, ascorbate peroxidase; CAT, catalase; MDA, malondialdehyde; PAL, phenylalanine ammonia lyase; POD, peroxidase; PPO, polyphenol oxidase; SP, soluble protein; SS, soluble sugar.

**Figure 6 ijms-24-08138-f006:**
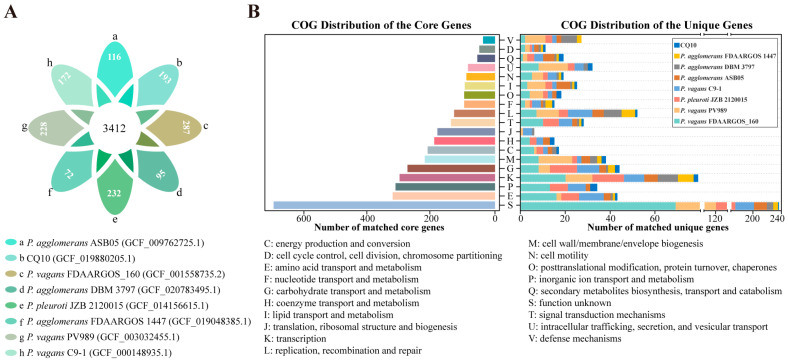
Pan–genome analysis of the strain CQ10 and related *Pantoea* species. (**A**) Petal diagram of orthologous genes of eight *Pantoea* strains. The number of core genes in eight *Pantoea* strains were represented in the center and unique genes (strain–specific genes) were in the petals. Latest Refseq accession numbers for strains are indicated in the legend in parentheses. (**B**) COG functional distribution for core and unique genes of eight *Pantoea* strains. The capital letters on the vertical axis indicate the COG categories.

**Figure 7 ijms-24-08138-f007:**
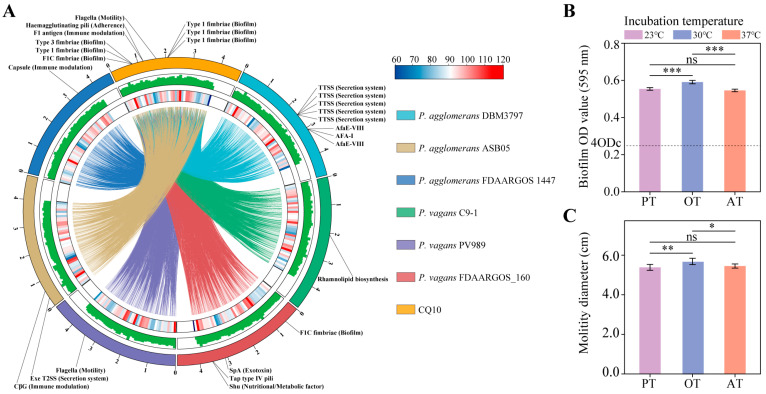
Virulence factor analysis of strain CQ10. (**A**) Circos representation of strain CQ10 compared with six *Pantoea* species. From outside to the center: (1) unique virulence factors located on chromosomes of each strain; (2) size (Mb) of the assembly for each genome; (3) bar chart and heatmap of the density distribution of genes; (4) central colored lines represent syntenic links between the chromosomes. (**B**) Biofilm formation of strain CQ10. (**C**) Motility of strain CQ10. PT, the temperature during strain CQ10 and plant interaction. OT, optimum temperature for strain growth in medium. AT, the temperature during strain CQ10 and animal interaction. Statistical analysis was performed with one–way ANOVA, and Tukey post–test. *n* = 6; mean ± SD. *, *p* < 0.05; **, *p* < 0.01; ***, *p* < 0.001; ns, not significant.

**Figure 8 ijms-24-08138-f008:**
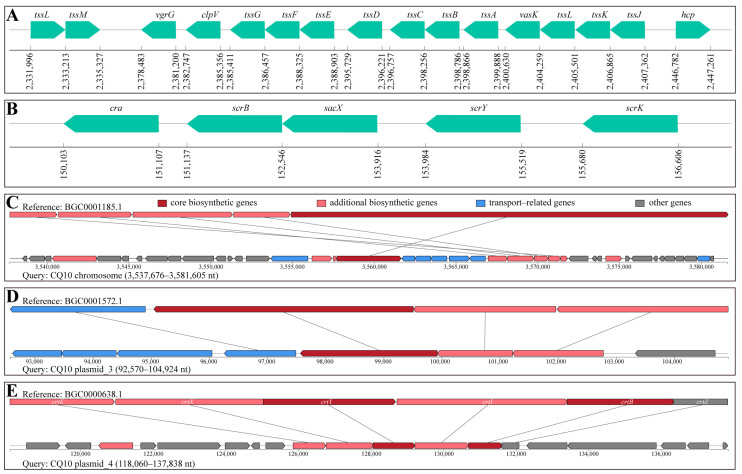
Arrangement of genes involved in the infection process in strain CQ10 genome. Schematic representation of the gene cluster coding for the (**A**) T6SS located on the chromosome and (**B**) sucrose metabolism located on a plasmid. Genes have been assigned identities based on their gene names or possible gene products. Comparison of the gene arrangement of the (**C**) bacillibactin, (**D**) desferrioxamine E, (**E**) carotenoid biosynthesis gene clusters between the strain CQ10 and the closest reference sequences in the minimum information about a biosynthetic gene cluster (MIBiG) database.

**Table 1 ijms-24-08138-t001:** Characteristics that distinguish the novel species from closest *Pantoea* species.

Characteristic	1	2	3	4
Utilization of				
Amygdalina	+	−	−	+
L–fucose	+	−	−	+
L–pyroglutamic acid	+	−	−	−
Tween 40	−	−	+	+
D–fucose	+	−	+	+
D–arabitol	+	+	−	+
Raffinose	−	−	−	+

Species: 1, *P. alfalfae* sp. nov. CQ10; 2, *P. agglomerans* JCM 1236; 3, *P. vagans* DSM 23078; 4, *P. brenneri* LMG 5343. +, positive; −, negative.

## Data Availability

The datasets presented in this study can be found in the main text and Supporting Information of this article. Genomic data of strain CQ10 can be found in the NCBI database (accession number CP082292.1–CP082300.1).

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
