# Peer review of "Diverse Virulence Attributes of Pantoea alfalfae sp. nov. CQ10 Responsible for Bacterial Leaf Blight in Alfalfa Revealed by Genomic Analysis"

_ijms, 2023, doi:10.3390/ijms24098138_

Round 1

Reviewer 1 Report

Comments to the authors:

In this study, the authors isolate a new strain of Pantoea that was present on the interior of alfalfa seeds. The strain was sequenced and a comparative genomic analysis was completed which identified unique virulence factors associated with motility and biofilm formation. A number of experiments confirmed the role of Pantoea alfalfa CQ10 in causing leaf blight in alfalfa and the involvement of a number of key bacterial physiological outputs associated with virulence including motility, biofilm formation, stress enzymes, Type VI secretion, nutrient metabolism and iron acquisition.  The following comments are offered to the authors to further improve the manuscript.

1. Lines 176, 177 and 190:  All of the three letter designations for the various enzyme assays should be defined in the main text, not just the figure legend.  In addition, a brief explanation of the importance to these factors in bacterial physiology and virulence would be beneficial for the reader.

2.  Figure 4: The indicators of statistical significance (*) have not been consistently displayed here.  Sometime they are over the CK data and sometimes they are over the CQ10 data making it hard to correlate the text and the figure.  The statistical analysis associated with this figure should be redone so that all panels reflect the statistical comparisons were performed in the same manner across all samples.

3.  Figure 7BC:  It is very difficult to see the statistical analysis indicators in these panels due to the small font size. The abbreviation for not significant (ns?) should be defined.

4.  Line 376:  It is unclear what is meant by the strain name being inaccurate.  A request to correct this information in the public database should be made.

5.  Line 638:  The accession number for the genomic sequence information should be stated as part of the Data Availability Statement.

6. Minor errors in writing:

Line 28:  Change “legume forage” to “forage legume”

Line 32: Remove “The”, change to Bacterial disease

Line 141:  Add “plants”, CQ-10-treated plants

Line 373:  Add “the”, in the genus

Reviewer 2 Report

-The authors report the isolation of an endophytic bacterial strain CQ10 from seeds of alfalfa. The whole-genome of the strain was sequenced and analyzed and the authors conclude that this strain is a new species within the genus Pantoea and proposed the name Pantoea alfalfae.

-From a taxonomic view point, strain CQ10 was not adequately compare the strain to all TYPE STRAINS of validly published Pantoea species: (Pantoea ananatis, Pantoea anthophila, Pantoea brenneri, Pantoea conspicua, Pantoea cypripedii, Pantoea deleyi, Pantoea dispersa, Pantoea eucalypti, Pantoea eucrina, Pantoea gaviniae, Pantoea piersonii, Pantoea punctata, Pantoea rodasii, Pantoea rwandensis, Pantoea septica, Pantoea stewartia, Pantoea vagans, and Pantoea wallisii).  The authors should consider use the 16S rRNA strains of the type strains of these species to generate a new phylogenetic tree to replace Figure S3.  The authors used some of these species in Figure S3 but all are required.

-In addition, Figure S3 which presented a phylogenetic tree based on 16S rRNA tree showed Pantoea ananatis LMG 2665 and Pantoea ananatis NCPPB 1846, which are the same species, clustering separately. Also, It is unclear why the author used in Figure S3 strains NCTC 9381, DSM 3493, NBRC 102470 and LMG 1286. These strains represent the same type strain, Pantoea agglomerans, in different culture collections, UK, Germany,   Japan and Belgium, respectively. This suggests that the authors probably might need to include a seasoned taxonomist to help with this aspect of the manuscript.

-Also, for the phenotypic characterization (Table 1), the authors used API 20E and Biolog GenIII systems (Lines 484-485) to generate their corresponded data  and indicated using the previously generated data of Gavini et al. [27] and Brady et al. [28] for P. agglomerans  and P. vagans, respectively (Lines 132-133). But the neither Gavini et al. [27] and Brady et al. [28] used GEN III (Biolog) which makes the data non-comparable.  Brady et al. used the biolog GN system.  The authors should consider redoing the phenotypical data, in parallel, and based on the current Figure S3, to include Pantoea agglomerans, Pantoea vagans  and Pantoea brenneri. Also, Table 1 does not indicate the strains of Pantoea agglomerans, Pantoea vagans that were used.

-The authors analyzed the whole genome and identified some virulence genes/proteins which they claim/speculate are responsible for bacterial leaf blight of alfalfa (Title) without knockout experiments. Also, the authors compared the whole-genome sequence of CQ10 to seven Pantoea strains of the species P. agglomerans (2 strains), P. vagans (3 strains) and P. pleuroti  (1 strain) (Figure 6 and Figure 7). It is not clear why the authors selected these strains. The authors should consider using only the closest type strains.

-The authors demonstrated that strain CQ10 is pathogenic to alfalfa seedlings and studied changes in key enzymes, 21 days after inoculation. At 21 days, the CQ10-inoculated plants  are “practically" dead (based on the images). A fundamental question is what meaningful enzymatic activity dead  can be reliably generated from dead/wilted plants? Why was data not collected at different time intervals, e.g. 0, 7, 14 and 21 days after inoculation? This might be an extreme suggestion that the authors consider deleting the sections on enzymatic activity. Or re-do the experiment and collect data at at least four time intervals after inoculation.

Round 2

Reviewer 2 Report

The authors have addressed most of the concerns. However, additional work is required on the phenotypic characteristics that differentiate the proposed species from closest validly published species. Phenotypic characteristics are one of the key data points in the description of new bacterial species. As such appropriate/reliable data are needed. The reviewer is well aware of the transition from Biolog GN to GN2 to GENIII and read the email the authors received from Biolog. The email supporting the transition from GN2 to GENIII is not affirming that it is scientifically sound to collate data in this way. For example, lines 133-135 “The phenotypic datas of P. agglomerans, P. vagans, and P. brenneri were obtained from Gavini et al. [27], Grimont and Grimont [28], and Brady et al. [29] and [30], respectively.” suggest that data of these three Pantoea species (Table 1) were obtained from four previously published documents. It is unclear which data came from which article. Gavini et al. used only API 20 E in their study. Brady et al. (2009) used API 20E, API 50CHB/E and Biotype-100 strips (bioMe ́rieux) as well as Biolog GN. Brady et al. (2010) used API 50CHB/E and Biotype-100 and used API 50 CHB/E (bioMe ́rieux) and GN2 MicroPlate (Biolog) for all Pectobacterium cypripedii strains. Grimont and Grimont [28] is a book chapter in Bergey’s Manual and the methods used to gather the data presented in Table BXII.g.240 could not be found. It possible these data is from other publications. However, these numerous sources are confusing. Based on the facts indicated above, the reviewer is suggesting that the authors re-do, in parallel (at the same time), the phenotypic assays for strain CQ10 and the three type strains of  Pantoea to improve on the reliability of the data.

Round 3

Reviewer 2 Report

The authors have re-determined the phenotypic data using API 20 E and GENIII, in parallel, with the required type strains. Biolog Inc is a business entity and does not have scientific authority on data reliability in the description of new bacterial taxon/taxa.